# Computational Design of Ni_6_@Pt_1_M_31_ Clusters for Multifunctional Electrocatalysts

**DOI:** 10.3390/molecules28227563

**Published:** 2023-11-13

**Authors:** Jiaojiao Jia, Dongxu Tian

**Affiliations:** School of Chemistry, Dalian University of Technology, No. 2 Linggong Road, Dalian 116024, China; jiajj0102@163.com

**Keywords:** DFT, single-atom catalyst, core–shell, HER, ORR, OER

## Abstract

High-efficiency and low-cost multifunctional electrocatalysts for hydrogen evolution reaction (HERs), oxygen evolution reaction (OERs) and oxygen reduction reaction (ORRs) are important for the practical applications of regenerative fuel cells. The activity trends of core–shell Ni_6_@M_32_ and Ni_6_@Pt1M31 (M = Pt, Pd, Cu, Ag, Au) were investigated using the density functional theory (DFT). Rate constant calculations indicated that Ni_6_@Pt_1_Ag_31_ was an efficient HER catalyst. The Volmer–Tafel process was the kinetically favorable reaction pathway for Ni_6_@Pt_1_M_31_. The Volmer–Heyrovsky reaction mechanism was preferred for Ni_6_@M_32_. The Pt active site reduced the energy barrier and changed the reaction mechanism. The ORR and OER overpotentials of Ni_6_@Pt_1_Ag_31_ were calculated to be 0.12 and 0.33 V, indicating that Ni_6_@Pt_1_Ag_31_ could be a promising multifunctional electrocatalyst. Ni_6_@Pt_1_M_31_ core–shell clusters present abundant active sites with a moderate adsorption strength for *H, *O, *OH and *OOH. The present study shows that embedding a single Pt atom onto a Ni@M core–shell cluster is a rational strategy for designing an effective multifunctional electrocatalyst.

## 1. Introduction

Reversible fuel cells or unitized regenerative fuel cells (URFCs) combine the functions of fuel cells and electrolyzers, which attract increasing attention because of their high levels of efficiency and environmentally friendly merits [1]. The key challenge is to develop highly active and stable multifunctional electrocatalysts for hydrogen and oxygen electrode reactions [2,3]. The cathodic hydrogen evolution reaction (HER) and the anodic oxygen evolution reaction (OER) have sluggish kinetics due to large overpotentials. Pt/C and noble metal oxides of RuO_2_ and IrO_2_ demonstrated advanced activity with reduced overpotentials in ORR/HER/OER processes [4,5,6]. However, the wider applications of noble-metal materials have been limited by their high price and scarcity [7,8]. Thus, efforts are mainly devoted to the design of efficient and cheap catalysts whose catalytic performances are comparable to noble metals in accelerating the ORR, OER and HER [9,10,11,12,13].

Recently, electrocatalysts with core–shell structures were prepared to be applicable for catalyzing the ORR/OER/HER efficiently [14,15]. The electronic structure of a core–shell cluster can be regulated through the electronic interaction between the core and shell [16]. Many core–shell nanoclusters have been explored theoretically and experimentally for the ORR process [17,18,19]. A novel Pt core–shell catalyst for the ORR was reported with an impressive improvement, producing 3.7 times greater results relative to a Pt/C material [20]. Chen and colleagues reported that Ni-Pd alloys with Pd-enriched surfaces exhibited significantly enhanced ORR activities and improved CH_3_OH tolerances compared to Pd/C catalysts [21]. Lin et al. reported the synthesis of snap-bean-like, multi-dimensional core/shell Ni/NiCoP nano-heterojunctions (NHs), presenting improved stability and activity for both hydrogen evolution reactions (HERs) and oxygen evolution reactions (OERs) [22]. Ma et al. demonstrated a core–shell-structured non-noble-metal, Ni@In, loaded on silicon nanowire arrays (SiNWs) for the efficient reduction of CO_2_ to formate [23]. Park’s group synthesized a Ni-based core–shell material (Ni@Ni-NC) which displayed excellent electrocatalytic OER performance with over- and onset potentials of 371 mV and 1.51 V, respectively, which are superior to those of commercial IrO_2_ [24], and Yu’s group found that Ni@Pd core–shell nanoparticles exhibited robust ORR activity and stability in both acidic and alkaline electrolytes, comparable to commercially available Pt/C catalysts [25]. In addition, Lang’s group prepared binary core–shell Ni@Pt NPs [26] with good ORR performance and durability. A Ni_19_@Pt_70_ cluster exhibited better catalytic activity and stability for the ORR than Pt_79_ [27].

With nearly 100% atom utilization, a unique local coordination environment and electronic configuration, noble-metal single-atom catalysts (SACs) exhibit unique properties [28,29,30,31,32]. Li’s group predicted that the Pt_1_/PMA (phosphomolybdic acid cluster) catalyst is a promising multifunctional electrocatalyst for use in water splitting (*η*^HER^ = 0.02 V and *η*^OER^ = 0.49 V) and a metal–air battery (*η*^ORR^ = 0.79 V) [33]. Zhao et al. reported a single-atom Ru catalyst for the OER with the overpotential of 118 mV [34]. Zhang et al. proposed a single-atom Au electrocatalyst supported by NiFe-LDH with high activity for the OER [35].

A multifunctional electrocatalyst should have abundant active sites which remarkably facilitate the adsorption/desorption of *H, *O, *OH and *OOH intermediates and products or two reactions simultaneously. The adsorption and desorption energies of the reaction species should be moderate, according to the Sabatier’s principle [36].

We systematically investigated the stability and multifunctional catalytic behavior of Ni_6_@M_32_ (M = Pt, Pd, Cu, Ag, Au) and Ni_6_@Pt_1_M_31_ (M = Pd, Cu, Ag, Au) for the ORR, OER and HER using DFT calculations. The OER overpotential of Ni_6_@Pt_1_Ag_31_ (*η*^OER^ = 0.33 V) is lower than RuO_2_ (*η*^OER^ = 0.37 V) [34] and IrO_2_ (*η*^OER^ = 0.56 V) [37]. Rate constant calculations indicated that Ni_6_@Pt_1_Ag_31_ is an efficient HER catalyst. The Volmer–Tafel process was a kinetically favorable reaction pathway for Ni_6_@Pt_1_M_31_, while the Volmer–Heyrovsky process was the major pathway for Ni_6_@M_32_. The present study shows that embedding a single Pt atom onto core–shell Ni@PtM cluster is a rational strategy for designing a multifunctional electrocatalyst.

## 2. Results and Discussion

### 2.1. Structure and Stability Analysis

A core–shell Ni_6_@M_32_ (M = Pt, Pd, Cu, Ag, Au) cluster (Figure 1) was used as a catalyst model to understand the activity trends of the HER, OER and ORR. The average bond lengths were calculated and are shown in Appendix A. The average bond lengths of the Ni_6_@M_32_ (M = Pt, Pd, Cu, Ag, Au) clusters were shorter than those of pure M_32_ clusters arising from a lattice mismatch between the core and shell atoms. The average bond length of the shell decreases in the following order: Ni_6_@Au_32_ ≈ Ni_6_@Ag_32_ > Ni_6_@Pd_32_ > Ni_6_@Pt_32_ > Ni_6_@Cu_32_.

The formation energies of the Ni_6_@Pt_1_M_31_ structures were calculated according to Appendix A and are presented in Table 1. The negative values indicate that it is thermodynamically favorable for the formation of Ni_6_@Pt_1_M_31_ from Ni_6_@M_32_ and a Pt atom. The thermodynamic trend is Ni_6_@Pt_1_Pd_31_ (−0.56 eV) > Ni_6_@Pt_1_Ag_31_ (−0.51 eV) > Ni_6_@Pt_1_Cu_31_ (−0.08 eV) > Ni_6_@Pt_1_Au_31_ (0.33 eV).

The average binding energy was calculated according to Appendix A and is presented in Figure 2a. The average binding energy decreases in the following order: Ni_6_@Pt_32_ (4.53 eV) > Ni_6_@Pd_32_ (4.36 eV), Ni_6_@Pt_1_Pd_31_ (4.40 eV) > Ni_6_@Pt_1_Au_31_ (3.02 eV), Ni_6_@Au_32_ (2.96 eV) > Ni_6_@Pt_1_Ag_31_ (2.63 eV), Ni_6_@Ag_32_ (2.53 eV) > Ni_6_@Pt_1_Cu_31_ (0.42 eV), Ni_6_@Cu_32_ (0.25 eV). Ni_6_@Pt_32_ has the highest binding energy (4.53 eV) and is the most stable catalyst.

A Bader charge analysis was carried out to investigate the electronic properties of the Ni_6_@M_32_ and Ni_6_@Pt_1_M_31_. The electronegativities (Pauling scale) of Ni, Pt, Pd, Au, Ag and Cu are 1.91, 2.28, 2.20, 2.54, 1.93 and 1.90, respectively. As presented in Figure 2b, the charge transfer amount changes in the following order: Ni_6_@Pt_32_ > Ni_6_@Pd_32_, and Ni_6_@Au_32_ > Ni_6_@Ag_32_ > Ni_6_@Cu_32_. The charge transfer amount increases with the increased difference in the electronegativity between the core and shell elements. It is closely related to the average binding energy, as shown in Appendix A. The average binding energy increases with an increased amount of charge transferred between the core and shell.

The density of the states projected onto the d states of the Ni_6_@M_32_ clusters and Ni_6_@Pt_1_M_31_ (M = Pd, Cu, Ag, Au) clusters is shown in the Appendix A. The introduction of a Pt atom narrows the distribution of the d states. Its influence is smaller on the Ni_6_@Pd_32_ cluster than on the other three clusters. The Ni_6_@Pt_1_Ag_31_ cluster shows the most significant change in its PDOS compared to the Ni_6_@Ag_32_ cluster, with the population of high-energy states decreasing and the population of low-energy states increasing. This feature may present active sites with a moderate adsorption strength for *H, *O, *OH and *OOH.

### 2.2. Hydrogen Evolution Reaction (HER) Catalytic Activity

Mechanistically, the HER involves three possible principal steps in acidic media which occur via either the Volmer–Heyrovsky or Volmer–Tafel process as presented in Appendix A. The Volmer reaction refers to the initial adsorption of protons from the acid solution to form adsorbed H and is usually considered to be fast. The Heyrovsky reaction refers to the reaction of a proton in the water layer with an adsorbed hydrogen to form H_2_. Two adsorbed H atoms react to form H_2_, which is Tafel reaction process.

#### 2.2.1. The Hydrogen-Adsorption Gibbs Free Energies of Ni_6_@M_32_ and Ni_6_@Pt_1_M_31_

To assess the Volmer performance of the Ni_6_@M_32_ (M = Pt, Pd, Cu, Ag, Au) and Ni_6_@Pt_1_M_31_ (M = Pd, Cu, Ag, Au), we calculated the differential hydrogen-adsorption Gibbs free energy (d−Δ*G*_H*_) and average-hydrogen-adsorption Gibbs free energy (a−Δ*G*_H*_) values (Appendix A) as functions of the hydrogen coverage (*θ*_H*_). Here, *θ*_H*_ was defined as n/6, where n is the number of adsorbed H atoms. The exchange current density value *i*_0_ was calculated as a function of d−Δ*G*_H*_ according to the Appendix A. The d−Δ*G*_H*_ and a−Δ*G*_H*_ values of Ni_6_@Pt_1_M_31_ (M = Pd, Cu, Ag, Au) from *θ*_H*_ = 1/6 to 6/6 are presented in Figure 3a,b,d,e,g,h,j,k. A volcano relation formed between d−Δ*G*_H*_ and the exchange current density. The volcano plot of *i*_0_ as a function of d−Δ*G*_*H_ is presented in Figure 3c,f,i,l. For comparison, data regarding Ni_6_@M_32_ (M = Pt, Pd, Cu, Ag, Au), including the d−Δ*G*_H*_, a−Δ*G*_H*_ and *i*_0_ values, are presented in Appendix A.

For Ni_6_@M_32_ and Ni_6_@Pt_1_M_31_ (M = Pd, Pt), the high hydrogen coverage value (*θ*_H*_ = 6/6) corresponds to the optimal d−Δ*G*_H*_ and *i*_0_ values. For example, for Ni_6_@Pt_1_Pd_31_, the d−Δ*G*_H*_ values were calculated to be −0.28, −0.23, −0.25, −0.12, −0.33 and 0.10 eV at a H coverage ranging from 1/6 to 6/6, and the volcano plot indicates that the high *θ*_H*_ = 6/6 corresponds to the low d−Δ*G*_H*_ and high *i*_0_ values. Compared with the Ni_6_@Pd_32_ cluster (Appendix A), the d−Δ*G*_H*_ and a−Δ*G*_H*_ values of Ni_6_@Pt_1_Pd_31_ are closer to zero, demonstrating the benefit of the introduction of single-site Pt. For Ni_6_@M_32_ (M = Cu, Ag, Au) and Ni_6_@Pt_1_M_31_ (M = Ag, Au), the low hydrogen coverage (*θ*_H*_ = 1/6) value corresponds to the optimal d−Δ*G*_H*_ and high *i*_0_ values.

The repulsion interaction between neighboring negatively charged, adsorbed *H atoms possibly improves the desorption. At low H coverage, Ni_6_@M_32_ (M = Cu, Ag, Au) present d−Δ*G*_H*_ and *i*_0_ values which are comparable to Ni_6_@Pt_32_. The optimal |d−Δ*G*_H*_|, |a−Δ*G*_H*_| and *i*_0_ values and the Bader charges of the hydrogen atoms in the Ni_6_@M_32_ and Ni_6_@Pt_1_M_31_ systems are presented in Appendix A. These results show the complex influence of the hydrogen coverage on the interaction between the adsorbate and the adsorption site. An analysis of the projected density of states (PDOS, Appendix A) shows that high coverage can broaden the distribution of the d-DOS of the adsorption sites for the Ni_6_@Pd_32_ and Ni_6_@Pt_1_Pd_31_ clusters. This weakens the adsorption strength of the adsorbates at the adsorption site.

#### 2.2.2. The Energy Barriers of the Tafel and Heyrovsky Processes

From the point of the adsorption free energy, at a low H coverage, Ni_6_@M_32_ (M = Cu, Ag, Au) present Volmer activity comparable to Pt-based Ni_6_@Pt_32_. We further investigated the kinetic properties of all candidates. Considering the water environment, the energy barriers of the Tafel and Heyrovsky processes were calculated for Ni_6_@M_32_ and Ni_6_@Pt_1_M_31_, with a d−Δ*G*_*H_ value close to zero. Figure 4 displays the structures of the adsorption state, transition state and product species and the energy barriers. The energy barriers of the Tafel process for Ni_6_@Pt_1_M_31_ (M = Pd, Cu, Ag, Au) are 0.33, 0.21, 0.10 and 0.55 eV, respectively. We employed the H_3_O species to consider the H_3_O^+^ + e^−^ approximately for the simulation of the Heyrovsky step. The energy barriers of the Heyrovsky process were calculated to be, respectively, 0.89, 0.88, 0.45 and 0.77 eV. For Ni_6_@Pt_1_M_31_ (M = Pd, Cu, Au), the Heyrovsky process needs to overcome a relatively higher energy barrier to form H_2_ by H_3_O reacting with an adsorbed H*. The Volmer–Tafel reaction mechanism is main pathway for Ni_6_@Pt_1_M_31_ (M = Pd, Cu, Ag, Au) by combining two adsorbed hydrogens to form H_2_. The energy barriers of the Tafel process for Ni_6_@M_32_ (M = Pt, Pd, Cu, Ag) are 0.55, 0.66, 0.78 and 1.06 eV, respectively, as presented in Appendix A. The energy barriers of the Heyrovsky process for Ni_6_@M_32_ (M = Pt, Pd, Cu, Ag) are 0.43, 0.54, 0.14 and 0.70 eV, respectively. The Volmer–Heyrovsky reaction pathway is favorable for Ni_6_@M_32_ (M = Pt, Pd, Cu, Ag).

The rate constants (*k*/s) of the Tafel and Heyrovsky processes in the HER were calculated according to Appendix A, based on the Eyring transition state theory with Wigner correction [38], as presented in Table 2. By comparing the rate constants, the Volmer–Heyrovsky reaction mechanism was determined to be favorable for Ni_6_@M_32_, and the Volmer–Tafel process was determined to be a kinetically advantageous reaction pathway for Ni_6_@Pt_1_M_31_. The Pt active site decreases the energy barrier and modulates the reaction mechanism.

Platinum metal catalysts have been known as the most efficient catalysts for the HER due to their optimum Gibbs free energy for atomic hydrogen adsorption (Δ*G*_H*_). Ni_6_@Cu_32_ has comparable HER activity to Ni_6_@Pt_32_, and at *θ*_H*_ = 2/6, Ni_6_@Pt_1_Ag_31_ is the most efficient catalyst of Ni_6_@M_32_ and Ni_6_@Pt_1_M_31_. These results indicate the H coverage is an important factor in the kinetic activity of the HER.

### 2.3. The Oxygen Reduction Reaction (ORR)/Oxygen Evolution Reaction (OER): Catalytic Activity

The activity of the ORR and OER under acidic conditions were investigated for Ni_6_@M_32_ and Ni_6_@Pt_1_M_31_. It was given that the oxygen reduction reaction pathway occurs via an associative mechanism, with four protonation reaction steps. The adsorption free energies of the oxygenated intermediates were calculated as presented in Appendix A via the computational hydrogen electrode (CHE) model, using hydrogen and water as references, according to Appendix A. The free energy diagram of the four-electron reaction is presented in Figure 5. The OER/ORR overpotential was obtained by Appendix A. As shown in Figure 4, the ORR’s potential-limiting step is the proton-coupled electron transfer step *OH → H_2_O for Pt_38_, Pt (111), Ni_6_@Pt_1_Cu_31_ and Ni_6_@M_32_ (M = Pd, Cu, Ag, Au). The potential-limiting step of the ORR’s elementary reactions for Ni_6_@Pt_1_M_31_ (M = Pd, Au) is O_2_ + H^+^ + e^−^ → *OOH. For Ni_6_@Pt_32_ and Ni_6_@Pt_1_Ag_31_, *O → *OH is the potential-limiting step. The ORR overpotentials are presented in Appendix A. The overpotential (*η*^ORR^) values of the Pt (111) and Pt_38_ clusters were calculated to be 0.73 and 0.81 V. For Ni_6_@Pt_1_M_31_ (M = Pd, Cu, Ag, Au), the overpotentials were calculated to be 0.77, 0.83, 0.12 and 1.06 V, respectively. Ni_6_@Pt_1_Ag_31_ showed the best activity, with *η*^ORR^ = 0.12 V. In the OER process, the potential-limiting step is the *O → *OOH step for Pt_38_, Pt (111), Ni_6_@M_32_ (M = Pt, Pd) and Ni_6_@Pt_1_M_31_ (M = Pd, Cu, Au). For Ni_6_@Cu_32_ and Ni_6_@Pt_1_Ag_31_, the OER’s potential-limiting step is the *OOH → O_2_ process. The second deprotonation process (*OH → *O) is the OER’s potential-limiting step for Ni_6_@M_32_ (M = Ag, Au). The OER overpotentials for Pt_38_ and Pt (111) are 1.34 and 0.74 V. The OER overpotentials for the Ni_6_@M_32_ (M = Pt, Pd, Cu, Ag, and Au) clusters are 1.17, 0.81, 0.92, 0.93 and 1.08 V, respectively. The calculated OER overpotentials for Ni_6_@Pt_1_M_31_ (M = Pd, Cu, Ag, Au) are respective 1.36, 1.61, 0.33 and 0.61 V. The OER overpotential for a Ni_6_@Pt_1_Ag_31_ (*η*^OER^ = 0.33 V) single-atom catalyst is lower than those of RuO_2_ (*η*^OER^ = 0.37V) [37] and IrO_2_ (*η*^OER^ = 0.56V) [37].

The adsorption strength of the three intermediates, *OOH, *O and *OH, was analyzed to explore the activity trend of the Ni_6_@M_32_ and Ni_6_@Pt_1_M_32_ (M = Pd, Cu, Ag, Au). As shown in Figure 6a,b, there is a linear relationship between the adsorption free energy values of *OOH and *OH. The inverted volcano curve was obtained by plotting the adsorption free energy of *OH and the corresponding ORR overpotential, as presented in Figure 6c. On the left leg, the rate-limiting step is the *OH → H_2_O process due to the strong *OH adsorption strength. On the right-side, because of the weak adsorption for *OOH, the process of generating *OOH intermediates from oxygen becomes the limiting step. A similar volcano curve can be obtained by plotting the adsorption free energy of the *OOH and the ORR overpotential, and the details are presented in Appendix A.

The d-band centers (*ε*_d_) were calculated to explain the interaction of the adsorbates with the adsorption sites because the O* adsorption strength of a metal is closely correlated with its d-orbital levels [39]. The d-band centers of the adsorption sites of Ni_6_@M_32_ and Ni_6_@Pt_1_M_31_ were calculated to explain the interaction of the *O intermediate with the adsorption sites. As shown in Figure 7a, there is no linear relationship between the adsorption energy of the *O intermediate and the d-band center of the adsorption site. A small deformation of the geometry of the metal cluster was caused by O adsorption. The adsorption energy was thought to be the superposition of the metal cluster’s geometry deformation energy and the binding energy between the metal cluster and the O. The adsorption energy was decomposed as follows: Δ*E*_ads_ = Δ*E*_deformation_ + Δ*E*_binding_, where Δ*E*_deformation_ is the difference in the energy of the metal cluster before and after the O adsorption; Δ*E*_binding_ is the binding energy of the O on the metal cluster. A linear relationship between the binding energy and the d-band center is presented in Figure 7b. The adsorption strength of the adsorbed oxygen intermediate decreases with the downshift in *ε*_d_ to the Fermi energy level due to the d-p anti-bond orbital occupancy [39].

The PDOS was analyzed to explore the interaction between O* and the active site, as presented in Figure 7c. The blue and green colors represent the d state distributions of the adsorption sites, and the red part represents the p-orbital distribution of the adsorbed oxygen atom. In the PDOS diagrams of Ni_6_@M_32_ (M = Au, Ag) and Ni_6_@Pt_1_M_31_ (M = Au, Ag), the antibonding orbitals are occupied, resulting in weakened adsorption strength. On the contrary, the PDOS diagrams of Ni_6_@M_32_ (M = Pt, Pd) and Ni_6_@Pt_1_M_31_ (M = Pt, Pd) show that the antibonding orbital energy levels shift up over the Fermi energy level, leading to a relatively stronger d-p interaction. Comparing Ni_6_@M_32_ and Ni_6_@Pt_1_M_31_, the single Pt atom causes the d-band centers to up-shift to the Fermi energy level, resulting in a slightly stronger adsorption strength. Combined with the d-band center and PDOS analyses, a moderate O adsorption interaction significantly improved the potential-limiting step *O → *OH, with a reduced overpotential of *η*^ORR^ = 0.12 V. Among all the candidates, Ni_6_@Pt_1_Ag_31_ is a promising multifunctional electrocatalyst for splitting water (∆*G*_TS_^Heyrovsky^ = 0.10 eV and *η*^OER^ = 0.33 V) and has a low overpotential of *η*^ORR^ = 0.21 V.

In summary, the electronic properties of the adsorption sites can be manipulated by changing the chemical composition of the core and shell. The free energies and electronic structures of the core–shell Ni_6_@Pt_1_M_32_ nanoclusters can be modulated to offer improved performance in the electrocatalysis of the ORR, HER and OER. In the ORR, Ni_6_@Pt_1_Pd_31_ and Ni_6_@Pt_1_Ag_31_ exhibit lower overpotentials than Pt (111). In the HER, Ni_6_@Cu_32_ (0.14 eV), Ni_6_@Pt_1_Cu_31_ (0.21 eV), Ni_6_@Pt_1_Pd_31_ (0.33 eV) and Ni_6_@Pt_1_Ag_31_ (0.10 eV) are promising efficient electrocatalysts. In the OER, the overpotentials of Ni_6_@Pt_1_Ag_31_ and Ni_6_@Pt_1_Au_31_ are lowered compared to that of Pt (111). 

## 3. Computational Details

### 3.1. Computational Methods

Spin-polarized DFT methods were implemented in the Vienna Ab initio Simulation Package (VASP) [40,41,42] to carry out energy calculations and a geometry optimization. The interaction between valence and core electrons was described using the projector-augmented wave (PAW) pseudopotential [43,44], with an energy cutoff for the plane waves of 400 eV. A generalized gradient approximation (GGA) with the PW91 functional was used to calculate the exchange–correlation energies [45,46]. The clusters were placed at the center of a 20 × 20 × 20 Å^3^ cubic box, 65% of which was vacuum space to avoid interaction between the clusters and their images. A Pt (111) slab was modeled using a five-layer, periodically repeated √3 × √2 super cell with 35 Å of vacuum space. A Gamma (1 × 1 × 1) point mesh was used during the geometric optimization of clusters. A threshold for self-consistent calculations of 1 × 10^−5^ eV was used, and 0.02 eV/Å was used for ionic optimization. The transition state (TS) was determined via the Dimer method [47]. A Bader charge analysis was conducted to analyze the charge transfer [48].

### 3.2. Models

A truncated octahedral (TO) structure was employed as the theoretical model. The shell of the TO structure consists of eight (111) planes and six (100) planes. There are eight non-equivalent adsorption sites (two top sites, three bridge sites, two face sites and one hollow site) on the Pt_38_ and Ni_6_@M_32_ (M = Pt, Pd, Cu, Ag, Au) clusters. The structures are shown in Figure 1.

## 4. Conclusions

A high-efficiency and low-cost multifunctional electrocatalyst for splitting water and reducing oxygen is required for the practical applications of regenerative fuel cells. The HER, OER and ORR activity trends were investigated for core–shell Ni_6_@M_32_ (M = Pt, Pd, Cu, Ag, Au) and Ni_6_@Pt_1_M_31_ (M = Pd, Cu, Ag, Au) via first-principles calculations. The core–shell Ni_6_@Pt_1_M_31_ clusters present abundant active sites with moderate adsorption strengths for *H, *O, *OH and *OOH. The HER reaction energy barriers and rate constants indicated that Ni_6_@Pt_1_Ag_31_ was the most efficient catalyst of all the Ni_6_@M_32_ (M = Pt, Pd) and Ni_6_@Pt_1_M_31_ clusters at an H coverage of *θ*_H*_ = 2/6. The Pt active site decreased the energy barrier and changed the reaction mechanism. The Volmer–Heyrovsky reaction mechanism was favorable for Ni_6_@M_32_, and the Volmer–Tafel process was a kinetically advantageous reaction pathway for Ni_6_@Pt_1_M_31_. The ORR and OER overpotentials of Ni_6_@Pt_1_Ag_31_ were calculated to be 0.12 and 0.33 V, and the OER overpotential was lower than that of RuO_2_ and IrO_2_, which proves that Ni_6_@Pt_1_Ag_31_ is a promising multifunctional electrocatalyst for the OER, HER and ORR. The present work provides significant insights for further searches for a high-efficiency and low-cost multifunctional electrocatalyst for regenerative fuel cells. 

## Figures and Tables

**Figure 1 molecules-28-07563-f001:**
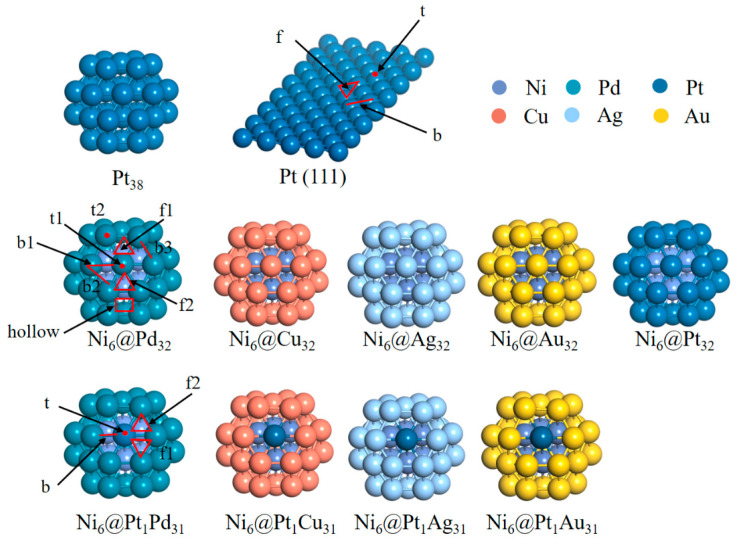
The structures of Pt_38_, Pt (111), core–shell Ni_6_@M_32_ (M = Pt, Pd, Cu, Ag, Au) and Ni_6_@Pt_1_M_31_ (M = Pd, Cu, Ag, Au) clusters. T, b, f and hollow represent the top, bridge, face and hollow site, respectively.

**Figure 2 molecules-28-07563-f002:**
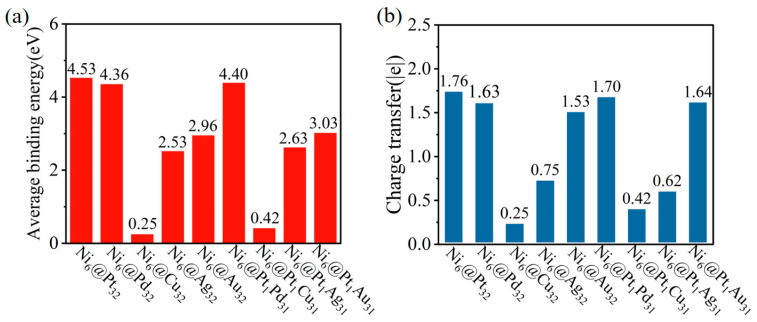
(**a**) Average binding energies for core–shell clusters. (**b**) Bader charge transfer amounts between the cores and shells of clusters.

**Figure 3 molecules-28-07563-f003:**
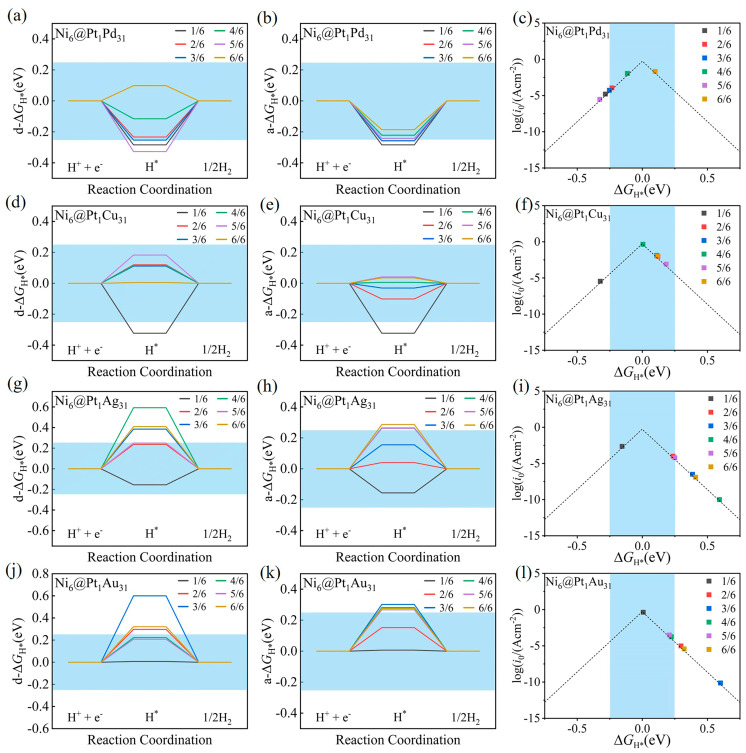
The differential Gibbs free energy profiles (d−Δ*G*_H*_) over the Ni_6_@Pt_1_M_31_ (M = Pd, Cu, Ag, Au) clusters (**a**,**d**,**g**,**j**); the average Gibbs free energy profiles (a−Δ*G*_H*_) (**b**,**e**,**h**,**k**); the volcano plot of *i*_0_ as a function of d−Δ*G*_H*_ at the best H coverage (**c**,**f**,**i**,**l**); the highlight in blue denotes the free energy window of ±0.25 eV.

**Figure 4 molecules-28-07563-f004:**
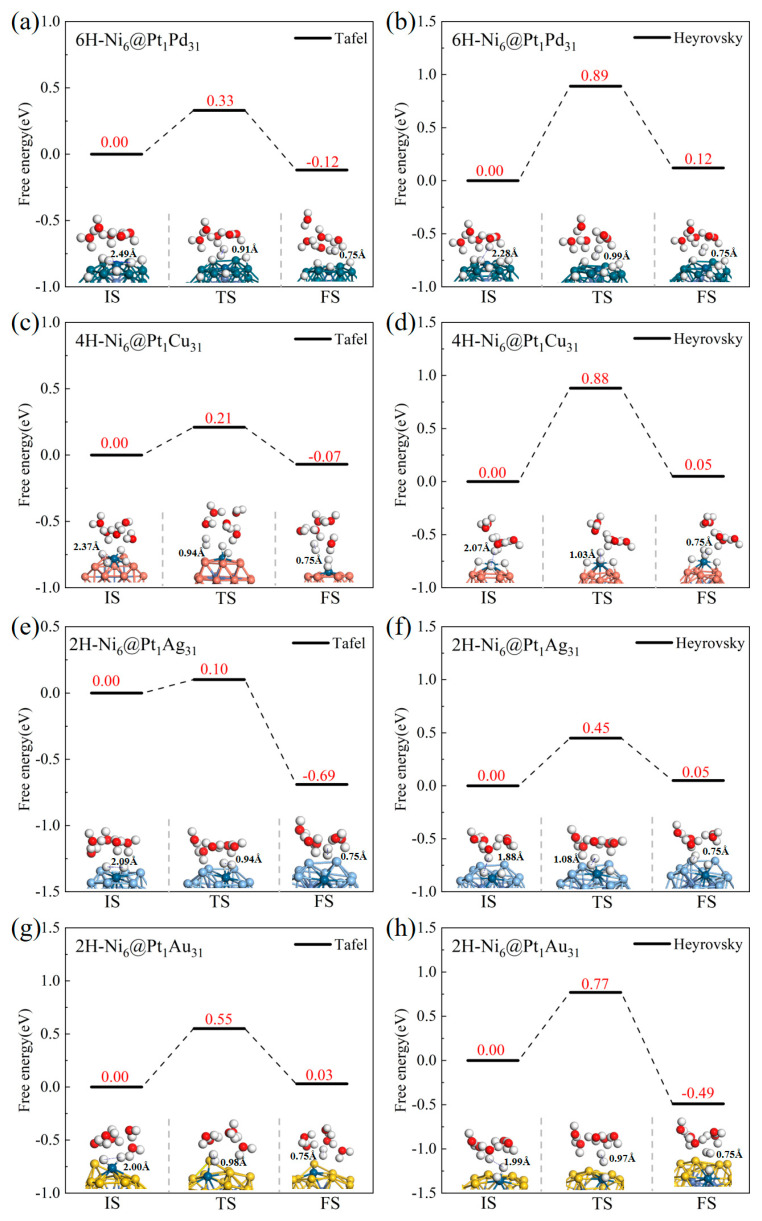
The potential energy profiles of the HER via the Tafel process and the Heyrovsky process at the optimal H coverage. 6H-Ni_6_@Pt_1_Pd_31_ (**a**,**b**), 4H-Ni_6_@Pt_1_Cu_31_ (**c**,**d**), 2H-Ni_6_@Pt_1_Ag_31_ (**e**,**f**) and 2H-Ni_6_@Pt_1_Au_31_ (**g**,**h**). White, red, blue, coppery, grey and golden ball represent H, O, Ni, Cu, Ag and Au, respectively.

**Figure 5 molecules-28-07563-f005:**
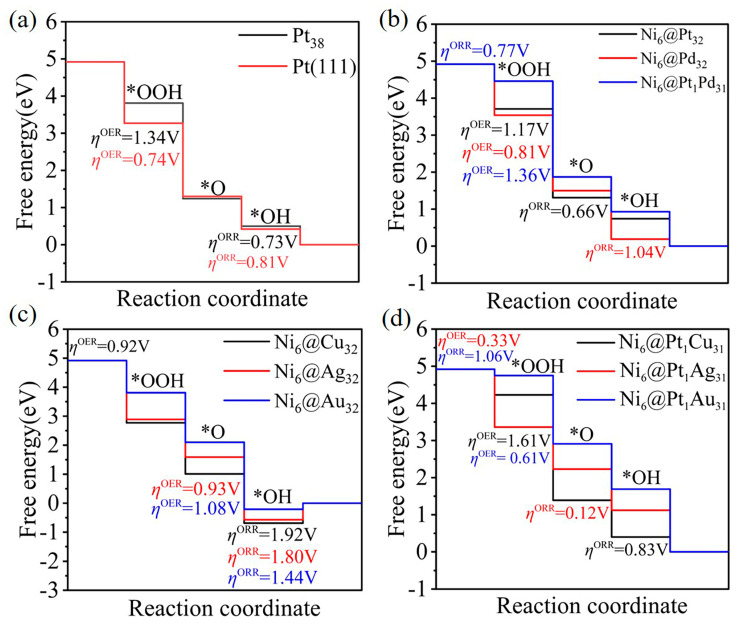
The Gibbs energy diagrams for the ORR and OER for (**a**) Pt_38_, Pt (111), (**b**) Ni_6_@M_32_ (M = Pt, Pd) and Ni_6_@Pt_1_Pd_31_, (**c**) Ni_6_@M_32_ (M = Cu, Ag, Au), (**d**) Ni_6_@Pt_1_Pd_31_ (M = Cu, Ag, Au) at U = 0 V (RHE).

**Figure 6 molecules-28-07563-f006:**
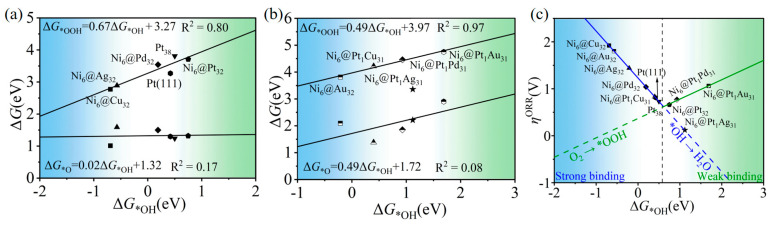
(**a**) Scaling relationships between the Gibbs free energy (Δ*G*) of *OOH and *OH for Ni_6_@M_32_ and (**b**) for Ni_6_@Pt_1_M_31_. (**c**) The linear relationship between the ORR overpotential and *OH adsorption free energy.

**Figure 7 molecules-28-07563-f007:**
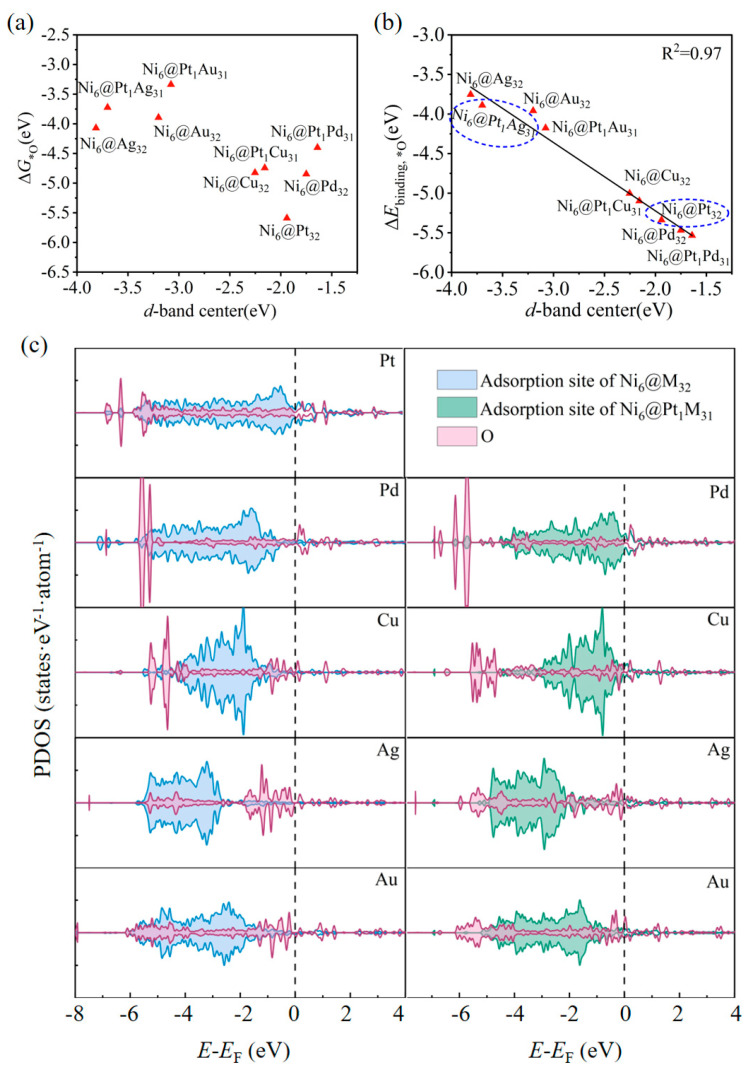
(**a**) Plot of the adsorption free energy of *O on Ni_6_@M_32_ and Ni_6_@Pt_1_M_31_ against the d-band center of the adsorption site. (**b**) Plot of the binding energy of *O on Ni_6_@M_32_ and Ni_6_@Pt_1_M_31_ against the d-band center of adsorption site. (**c**) Density of states projected onto the d states of adsorption site on *O of Ni_6_@M_32_ (blue), adsorption site on *O of Ni_6_@Pt_1_M_31_ (green), and the p states of adsorbed oxygen atoms, respectively (pink).

**Table 1 molecules-28-07563-t001:** The formation energies of the Ni_6_@Pt_1_M_31_ structures.

Reaction Coordination	*E*_f_/eV
Ni_6_@Pd_32_ + Pt_1_ → Ni_6_@Pt_1_Pd_31_ + Pd_1_	−0.56
Ni_6_@Cu_32_ + Pt_1_ → Ni_6_@Pt_1_Cu_31_ + Cu_1_	−0.08
Ni_6_@Ag_32_ + Pt_1_ → Ni_6_@Pt_1_Ag_31_ + Ag_1_	−0.51
Ni_6_@Au_32_ + Pt_1_ → Ni_6_@Pt_1_Au_31_ + Au_1_	0.33

**Table 2 molecules-28-07563-t002:** Rate constants (*k*/s) and energy barriers of the Tafel and Heyrovsky processes in the HER on Ni_6_@M_32_ (M = Pt, Pd, Cu, Ag) and Ni_6_@Pt_1_M_31_ (M = Pd, Cu, Ag, Au).

System	Δ*G*_Tafel_^≠^/eV	Tafel (*k*/s)	Δ*G*_Heyrovsky_^≠^/eV	Heyrovsky (*k*/s)
6H-Ni_6_@Pt_32_	0.55	4.35 × 10^3^	0.43	3.92 × 10^5^
6H-Ni_6_@Pd_32_	0.66	54.4	0.54	4.78 × 10^3^
2H-Ni_6_@Cu_32_	0.78	0.72	0.14	3.17 × 10^10^
3H-Ni_6_@Ag_32_	1.06	1.84 × 10^−5^	0.70	9.84
6H-Ni_6_@Pt_1_Pd_31_	0.33	1.76 × 10^7^	0.89	6.17 × 10^3^
4H-Ni_6_@Pt_1_Cu_31_	0.21	1.84 × 10^9^	0.88	9.25 × 10^−3^
2H-Ni_6_@Pt_1_Ag_31_	0.10	1.83 × 10^11^	0.45	1.69 × 10^5^
2H-Ni_6_@Pt_1_Au_31_	0.55	3.27 × 10^3^	0.77	0.64

## Data Availability

Data are contained within the article and Appendix A.

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
