# Peer review of "Computational Design of Ni6@Pt1M31 Clusters for Multifunctional Electrocatalysts"

_molecules, 2023, doi:10.3390/molecules28227563_

Round 1
Reviewer 1 Report
This article reports the results of a computational design of Ni6@Pt1M31 clusters for multifunctional electrocatalysts. The presented results are good and well-managed and citing the proper literature. However, some points need further improvements, and additional data should be supplied to improve the quality of the manuscript. Therefore, I recommend that this paper should be considered for publication after following major revision.
1. In the introduction part, the logicality is not very explicit, and there are some English grammatical errors; please double-check and correct them.
2. Authors are encouraged to cite some relevant and recent literature and improve the concept of the paper and highlight the novelty of the current work, such as.: Nano Energy 101 (2022): 107624., Chemical Engineering Journal, 440, 135928. Energy Storage Materials 45 (2022): 301-322. In order to strengthen their ideas in the light of previous work.
3. All equations should be checked and modified in math-type software.
4. It would be appreciated if the authors could draw a table for a better comparison in the main text.
5. The optimization and efficiency of the different kinds of species should be discussed in more detail.
6. The resolution of Fig. 2 and 4 should be enhanced.
Moderate English revision required.
Reviewer 2 Report
In this manuscript, the authors conducted a simulation experiment to design multifunctional electrocatalytic materials using the Ni6@M32 and Ni6@Pt1M31 (M = Pd, Cu, Ag, Au) design rules. While the attempted motivation of this work is commendable, the manuscript lacks fundamental insights and rationality in selecting this design strategy. Therefore, based on the current format, the manuscript cannot be accepted for publication in the Molecules journal. To enhance the manuscript's quality, a few comments are provided below:
What is the rationale behind choosing the Ni6@M32 parent structure? The role of Ni with 6 atoms per formula unit can be given in the manuscript.
If spin-polarized calculations are performed for some of the attempted dopants, would the authors obtain the same simulation results? Commenting on this would help expand the fundamental knowledge of the manuscript.
The used slab model can be included in the supporting information.
Is there a specific reason why only Gamma point calculation was used for relaxation?
If the adsorbate modifies the electronic structure of the catalytic surface through surface adsorption, how the PDOS will change according to different adsorption levels?
Any comments on defects in the cluster materials and their effect on the electronic structure can be given in the manuscript?
Minor editing is needed.
Reviewer 3 Report
See attached file.

Reviewer 4 Report
In this manuscript, the authors reported a density functional theory study of core-shell clusters of Ni6@M32 and Ni6@Pt1M31 for HER, ORR, and OER under the acid condition. The methodology has been well verified in other studies, with which the performance of the clusters in catalyzing the 4e-electron reaction path for the ORR and OER was evaluated, and the Ni6@Pt1Ag31 was identified to be a promising multifunctional electrocatalyst. The results enrich our understanding on the electrocatalysis of clusters with core-shell structure, and offer insights for the design of effective catalysts, thus the manuscript is recommended to be accepted for publication.
Below are some suggestions for the reference of the authors:
1. The authors correlated the order of charge transfer in Ni6@M32 and Ni6@Pt1M31 cluster to the electronegativity difference. From the statement in line 150, it also shows dependence on the group of the elements. The authors may think whether it is necessary to make a short statement on this point.
2. The authors introduced the so-called "differential hydrogen adsorption Gibbs free energy " to correlate the exchange current density. In an approach proposed by Norskov et al. (Journal of The Electrochemical Society, 2005, 152(3) J23-J26), the exchange current density was estimated according to the average hydrogen adsorption Gibbs free energy. Can the authors briefly address the necessity to introduce the differential hydrogen adsorption gibbs free energy?
3. How was the hydrogen coverage defined in the paper?
4. There are some grammar problems to be corrected, e.g.
Line 13: delete the word “While”;
Line 14: in the sentence of “The Pt active site”, the subject is in its single form, while the verbs (decrease, modulate) are in their plural forms. This should be corrected.
Line 153: delete “the”.
Round 2
Reviewer 1 Report
Accept
Moderate English revision is suggested.